# Visible to mid-infrared giant in-plane optical anisotropy in ternary van der Waals crystals

Yanze Feng[1,2,7], Runkun Chen[3,4,7], Junbo He[5,7], Liujian Qi[1,2], Yanan Zhang[1,2], Tian Sun[3], Xudan Zhu[5], Weiming Liu[5], Weiliang Ma[3], Wanfu Shen[6], Chunguang Hu[6], Xiaojuan Sun[1,2], Dabing Li[1,2]✉, Rongjun Zhang[5]✉, Peining Li[3]✉ & Shaojuan Li[1,2]✉

Birefringence is at the heart of photonic applications. Layered van der Waals materials inherently support considerable out-of-plane birefringence. However, funnelling light into their small nanoscale area parallel to its out-of-plane optical axis remains challenging. Thus far, the lack of large in-plane birefringence has been a major roadblock hindering their applications. Here, we introduce the presence of broadband, low-loss, giant birefringence in a biaxial van der Waals materials $Ta_2NiS_5$, spanning an ultrawide-band from visible to mid-infrared wavelengths of 0.3–16 μm. The in-plane birefringence $\Delta n \approx 2$ and 0.5 in the visible and mid-infrared ranges is one of the highest among van der Waals materials known to date. Meanwhile, the real-space propagating waveguide modes in $Ta_2NiS_5$ show strong in-plane anisotropy with a long propagation length (>20 μm) in the mid-infrared range. Our work may promote next-generation broadband and ultracompact integrated photonics based on van der Waals materials.

Optical anisotropy holds great promise for modern photonics by virtue of supporting various physical effects and important applications, including polarization control[1–4], ultra-confined light coupling[5,6], nonlinear and quantum optics[7–9]. The large birefringence parameter $\Delta n$ ($n$ is the refractive index), which indicates the difference in the speed of light between two crystallographic axes, is essential in achieving these important applications. Despite large birefringence is very essential, the currently commercial birefringence crystals such as $YVO_4$[10], $MgF_2$[11] and $CaCO_3$[12] have relatively small birefringence, with values below 0.2[13]. Some liquid crystals have shown an increase in birefringence, but typically fall below 0.4[14]. The uncovering of the quasi-one-dimensional crystal $BaTiS_3$ raises the record of the birefringence in nature to 0.76[15].

Although continuous efforts have been devoted to explore large birefringent materials, recently reported values in crystalline borates such as $Ba_3Mg_3(BO_3)_3F_3$[16], $K_5Ba_2(B_{10}O_{17})_2(BO_2)$[17] and $LiBO_2$[18] are still moderate. By artificially manipulating the crystal structure, researchers have achieved a colossal birefringence in $Sr_{9/8}TiS_3$ (-2.1)[19]. However, the relatively large volume of these bulk materials is not applicable to compact integrated photonic applications. The latest researches have suggested layered van der Waals (vdW) materials exhibit inherent large birefringence, as the weak interlayer bonding leads naturally to considerable out-of-plane birefringence (e.g., $MoS_2$ (-3)[20,21], $WS_2$ (-2)[22] and hBN (-0.7)[23]). Compared to the bulk crystals, layered vdW materials highlight with atomic flat smooth surfaces, strong light-matter

[1]State Key Laboratory of Luminescence and Applications, Changchun Institute of Optics, Fine Mechanics and Physics, Chinese Academy of Sciences, Changchun, Jilin 130033, China. [2]University of Chinese Academy of Sciences (UCAS), Beijing 100049, China. [3]Wuhan National Laboratory for Optoelectronics & School of Optical and Electronic Information, Huazhong University of Science and Technology, Wuhan 430074, China. [4]State Key Laboratory of Structural Chemistry, Fujian Institute of Research on the Structure of Matter, Chinese Academy of Sciences, Fuzhou 350002, China. [5]Department of Optical Science and Engineering, Shanghai Frontiers Science Research Base of Intelligent Optoelectronics and Proception, Institute of Optoelectronics, Fudan University, Shanghai 200433, China. [6]State Key Laboratory of Precision Measuring Technology and Instruments, Tianjin University, Weijin Road 92, Nankai District, Tianjin 300072, China. [7]These authors contributed equally: Yanze Feng, Runkun Chen, Junbo He. ✉e-mail: lidb@ciomp.ac.cn; rjzhang@fudan.edu.cn; lipn@hust.edu.cn; lishaojuan@ciomp.ac.cn

interactions, good flexibility and compatibility with the current silicon photonic technology[24–27], which makes them the promising candidates for next generation on-chip compact nanophotonic applications[28,29]. However, the optic axis of these layered materials is typically out of the plane, and thus their utility in conventional optical systems is limited by the difficulty to funnel light into the small nanoscale area that parallels to its out-of-plane optical axis. Instead, in-plane birefringence is more conducive to the practical applications[2,30], but the realization of large in-plane birefringence in these layered materials remains challenging.

Currently, vdW materials explored with in-plane optical anisotropy such as FePS₃[30], α-phase molybdenum trioxide (α-MoO₃)[31], rhenium diselenide (ReSe₂)[32], and black phosphorus[32] display a low in-plane birefringence. Achieving large in-plane optical birefringence is expected to require much larger structural anisotropy. Among various vdW materials with anisotropic structure, a ternary biaxial crystal Ta₂NiS₅ has reignited research interests recently[33–35]. This anisotropic crystal possesses layered orthorhombic structures, in which tetrahedral NiS₄ and octahedral TaS₆ units run along different in-plane directions, naturally revealing a distinct in-plane structural anisotropy, which is larger than the most investigated vdW materials (Supplementary Table 1). Particular attention should also be given to the different effective electronic polarizability along two in-plane principal directions of Ta₂NiS₅ in which $Ta^{4+}$ ions along the two axes is almost identically distributed, whereas $Ni^{2+}$ and $S^{2-}$ are preferentially distributed along $a$-axis and $c$-axis, respectively. As is known, the electronic polarizability of $Ni^{2+}$ ($1.107 Å^3$) is almost the lowest among bivalent metal ions[36], and $S^{2-}$ ($10.2 Å^3$) is much higher[15]. This distinct polarizability difference would cause a large anisotropy in optical

property between the $a$-axis and $c$-axis[15], providing a clue for the large in-plane birefringence in Ta₂NiS₅.

Here, we propose and experimentally demonstrate the presence of both significant in-plane and out-of-plane optical anisotropy in Ta₂NiS₅, spanning the ultrawide-band from visible to mid-infrared (MIR) range (0.3–16 μm). Specifically, we find the in-plane birefringence $\Delta n \approx 2$ and 0.5 in the visible and MIR ranges is one of the highest among vdW materials known to date. Meanwhile, the superior out-of-plane birefringence ($\Delta n > 1.6$) in the MIR region makes Ta₂NiS₅ as one of the excellent anisotropic vdW materials. Furthermore, combining with the near-field real-space nanoimaging, we reveal that the large refractive index, in-plane birefringence, and low extinction coefficient of Ta₂NiS₅ enable it a promising in-plane anisotropic and low-loss dielectric waveguide across an ultrabroadband range from 633 nm to 11.111 μm. The large optical anisotropy exhibited by Ta₂NiS₅ makes it a strong contender for advancing the pioneering field of nanophotonic and integrated optics and enabling an unique platform for the study of optical anisotropy.

## Results

### Giant birefringence of Ta₂NiS₅

Figure 1a illustrates the crystal structure of Ta₂NiS₅, a ternary vdW material. Ta₂NiS₅ has layered orthorhombic structures where Ni and Ta atoms link with the surrounding S atoms to form tetrahedral NiS₄ and octahedral TaS₆ units within the layer plane. These units run along different crystallographic directions with armchair or zigzag chain arrangements, respectively[33,34]. In light of the distinct chain structures along the $a$-axis and $c$-axis, considerable in-plane anisotropic optical properties could be expected in Ta₂NiS₅. The crystallographic

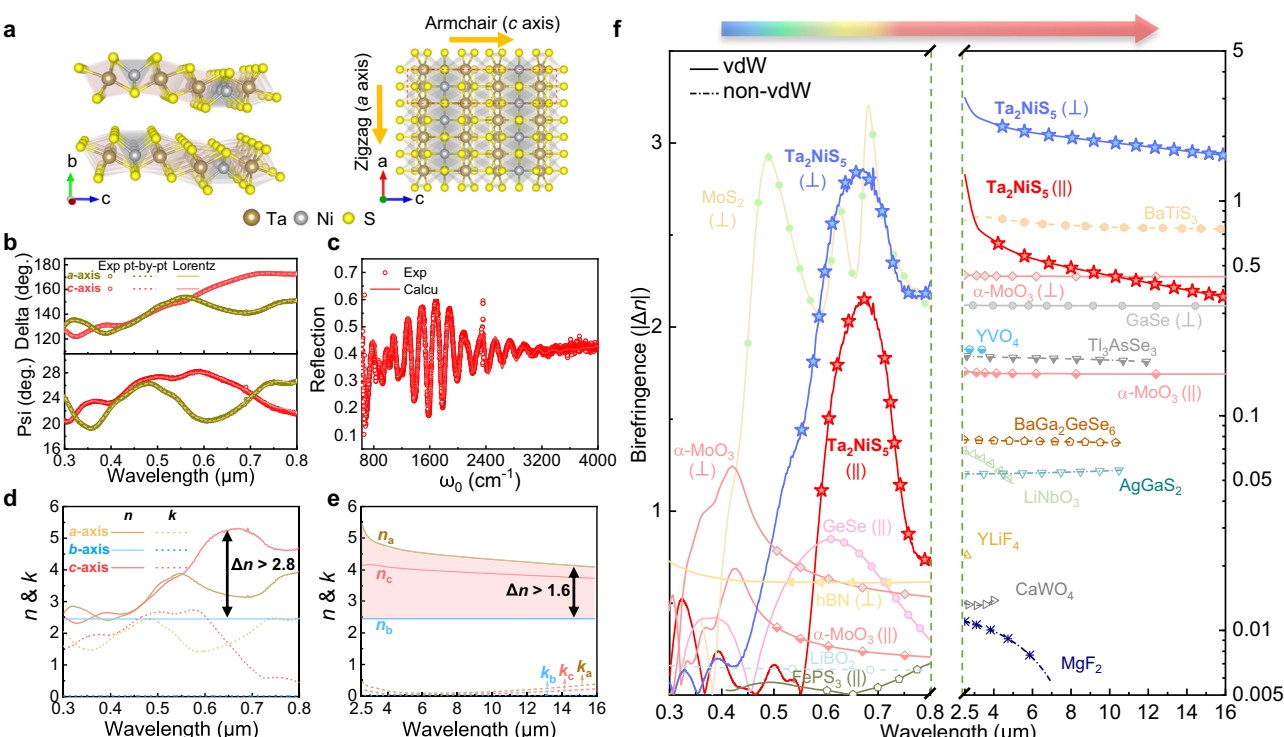

**Fig. 1 | Giant and ultra-broadband birefringence in Ta₂NiS₅ crystal. a** Crystal structures of Ta₂NiS₅. The red dash region represents the unit cell of Ta₂NiS₅. **b** The experimental (unfilled symbols) and fitted (dashed lines for point-by-point method; solid lines for Lorentz method) data of spectroscopic ellipsometry parameters Psi and Delta along the $a$-axis and $c$-axis of Ta₂NiS₅ flake. **c** The Fourier transform infrared spectroscopy (FTIR) experimental (unfilled symbols) and calculated (red line) reflectance spectra of a 5.58-μm-thick Ta₂NiS₅ sample on CaF₂ substrate. **d, e** Extracted complex refractive-index values of Ta₂NiS₅ for the $a$-, $b$-, and $c$-axis in

the visible and mid-infrared (MIR) range. The material shows a birefringence value as large as $\Delta n \approx 2.8$ in the visible and persists $\Delta n > 1.6$ in the MIR range (red shaded region). **f** Comparison of the absolute birefringence values of Ta₂NiS₅ with other anisotropic materials from previous works in the literature[10,11,15,18,20,23,30,31,45,62–68]. ‖ represents in-plane birefringence of vdW crystal, ⊥ represents out-of-plane birefringence of vdW crystal. The measurement wavelength range is currently limited by the capability of our equipment.

orientation of the exfoliated $Ta_2NiS_5$ flakes was determined by angle-resolved polarized Raman spectroscopy. The results reveal that the long-axis of the exfoliated flakes was aligned with the $a$-axis of the $Ta_2NiS_5$ crystal (Supplementary Note 1).

The in-plane optical constant of the $Ta_2NiS_5$ crystal in the visible range was measured by the spectroscopic ellipsometry. Spectroscopic ellipsometry has been widely used in recent years to investigate the optical properties of vdW materials[37,38]. To overcome the limitation of the lateral dimensions of the mechanically exfoliated nanosheets, the spectroscopic ellipsometer was coupled with a focused beam of spot size as small as $60 \times 25\,\mu m^2$. Here, we focus on investigating the in-plane ($ac$-plane) anisotropy of $Ta_2NiS_5$ nanosheets by rotating the in-plane azimuth angles of the samples (*see Methods*). The measured ellipsometric parameters Psi ($\Psi$) and Delta ($\Delta$) show pronounced differences specialized in the range of 600–800 nm (Fig. 1b), clearly indicating the large optical anisotropy. To extract the complex refractive indices from the measured Psi and Delta data, a four-phase model consisting of the $Si/SiO_2/Ta_2NiS_5$/air is considered (Supplementary Note 2). The optical constants of Si and $SiO_2$ were taken from the literature[39] and were validated through a pre-test using spectroscopic ellipsometry on the pristine substrate. The in-plane refractive indices of $Ta_2NiS_5$ layer were firstly extracted by the point-by-point method[40], which is a mathematical inversion method that accurately derives complex refractive indices but requires the prior knowledge of the thickness taken from the atomic force microscope (AFM) measurements. To verify the Kramers-Kronig consistency of the point-by-point result, the multi-oscillator Lorentz model was also applied (Supplementary Note 2). Both methods fit the Psi and Delta curves very well (Fig. 1b). To extract the complex refractive indices in MIR range, Fourier transform infrared spectroscopy (FTIR) was performed on a $Ta_2NiS_5$ sample on $CaF_2$ substrate to obtain the reflection spectra from 2.5 μm to 16 μm (Fig. 1c). A genetic algorithm[41] was used for fitting the reflection spectrum (Supplementary Note 3). The fitted reflectance spectrum (red line in Fig. 1c) achieves good consistency with the experiment data. This good agreement was further verified in multiple flakes with the same model parameters (Supplementary Fig. 6).

The anisotropic refractive indices of $Ta_2NiS_5$ in visible and MIR range were then plotted in Fig. 1d, e, respectively. As expected, the in-plane refractive index of $Ta_2NiS_5$ exhibits significant anisotropy, especially when the wavelength is greater than ~580 nm. In the visible range, the dominant peak ~600 nm on the extinction coefficient $k$ along the $c$-axis is assumed to be the interband transition from valence band of Ni $3d$ orbitals to the conduction band of Ta $5d$ orbitals[42–44]. However, this peak is absent for crystallographic direction along the $a$-axis. Besides, the extinction coefficient $k$ in the MIR range is negligible. Worthy of attention is that the in-plane birefringence values of $\Delta n \approx 2$ and 0.5 in the visible and MIR ranges, which are much larger than other in-plane anisotropic vdW materials reported, such as $FePS_3$ (~0.2 at visible)[30], GeSe (0.85 at 620 nm)[45] and α-$MoO_3$ (~0.17 at MIR)[31] as shown in Fig. 1f. On the other hand, $Ta_2NiS_5$ also exhibits a significant out-of-plane birefringence $\Delta n \approx 2.8$ in visible range and $\Delta n > 1.6$ in the MIR range (the out-of-plane refractive indices were confirmed by the near-field experiment discussed in the following part). Compared with bulk birefringent crystals and vdW materials (Fig. 1f), $Ta_2NiS_5$ has giant in-plane and out-of-plane birefringence in both the visible and MIR region making it as one of the promising anisotropic vdW materials. Meanwhile, the in-plane dichroism of $Ta_2NiS_5$ also reaches a high value of ~2.0 in visible range, and 0.357 in the MIR range (Supplementary Note 4). However, one should also be aware of the non-negligible absorption induced losses in the visible spectrum may impose possible limitations on future device design. To further verify the anisotropic optical properties of $Ta_2NiS_5$, we have also performed density functional theory (DFT) calculations to reproduce the experimental values (Supplementary Note 5). The DFT results can well match the

experimental optical properties, further validating high optical anisotropy of $Ta_2NiS_5$.

## Real-space nanoimaging of anisotropic waveguide modes

Notably, the absolute values of refractive indices of $Ta_2NiS_5$ are very high, especially the in-plane components. These extracted large refractive indices and giant birefringence indicate its potential application in compact nanophononics. To verify this, we analyzed the anisotropic propagation of planar waveguide modes in $Ta_2NiS_5$ flakes, using a scattering-type scanning near-field optical microscope (s-SNOM)[46–48]. According to previous studies[21], the transverse electric (TE) and transverse magnetic (TM) waveguide modes along $a$-axis in $Ta_2NiS_5$ film can be introduced as:

$$\sqrt{n_c^2 k_0^2 - q_{wm}^2}\,d = \tan^{-1}\left(\frac{\sqrt{q_{wm}^2 - k_0^2 n_1^2}}{\sqrt{n_c^2 k_0^2 - q_{wm}^2}}\right) + \tan^{-1}\left(\frac{\sqrt{q_{wm}^2 - k_0^2 n_s^2}}{\sqrt{n_c^2 k_0^2 - q_{wm}^2}}\right) + m\pi \quad (1)$$

$$\sqrt{\frac{n_a^2}{n_b^2}(n_b^2 k_0^2 - q_{wm}^2)}\,d = \tan^{-1}\left(\frac{n_a^2 \sqrt{q_{wm}^2 - k_0^2 n_1^2}}{n_1^2 \sqrt{\frac{n_a^2}{n_b^2}(n_b^2 k_0^2 - q_{wm}^2)}}\right) + \tan^{-1}\left(\frac{n_a^2 \sqrt{q_{wm}^2 - k_0^2 n_s^2}}{n_s^2 \sqrt{\frac{n_a^2}{n_b^2}(n_b^2 k_0^2 - q_{wm}^2)}}\right) + n\pi \quad (2)$$

where $k_O$ represents the incident wavevector in free space, $q_{wm}$ represents the wavevectors of in-plane propagation waveguide mode, $n_1$ and $n_s$ correspond to the refractive indices of air and substrate respectively, $n_a$, $n_c$ and $n_b$ correspond to the tensor of refractive indices of the $Ta_2NiS_5$ crystal along different crystallographic direction, $m$ and $n$ are the order number of $TE_m$ and $TM_n$, respectively. Apparently, the wavevectors of TE and TM mode are related with the tensor of refractive indices, which provides an extra way to verify the extracted large optical anisotropy from the far-field experiments.

The schematics of the near-field experiment was shown in Fig. 2a. The $Ta_2NiS_5$ flakes were exfoliated onto standard silicon wafer with 290-nm-thick $SiO_2$. The angle of the incident light ($k_0$) relative to the sample surface (x−y plane) is defined as $\alpha$ ($\alpha$ is fixed at 30° in our experiments), and the angle between the projection of the incident light in the x−y plane ($k_{xy}$) and the sample's edge ($a$-axis) is defined as $\beta$. The $p$-polarized laser beam with different excitation wavelengths is focused onto the apex of the AFM tip to excite waveguide modes in the $Ta_2NiS_5$ flakes. The excited waveguide modes then propagate in the $Ta_2NiS_5$ flakes as cylindrical waves and get scattered into the far-field as free-space light at the sample edges and interfere with the tip-scattered light at the detector. Since the in-plane wavevectors are closer to the free space wavevector than those of the graphene surface plasmon polaritons[49,50] and phonon polariton in hBN[51] and α-$MoO_3$[5], the waveguide modes tend to scatter into air thereby the back reflected portion is negligible and the relevant interference fringe pattern is not observed here (Supplementary Note 6). Instead, the tip-launched waveguide modes interfere with the tip scattered light and form interference fringe patterns depending on the incident angle. The relationship between different rotation angles, the interference fringe wavevectors $q_{obs}$, and the in-plane wavevectors $q_{wm}$ can be expressed as[52]

$$q_{obs} = q_{wm}\cos\gamma + k_0\sin\beta\cos\alpha \quad (3)$$

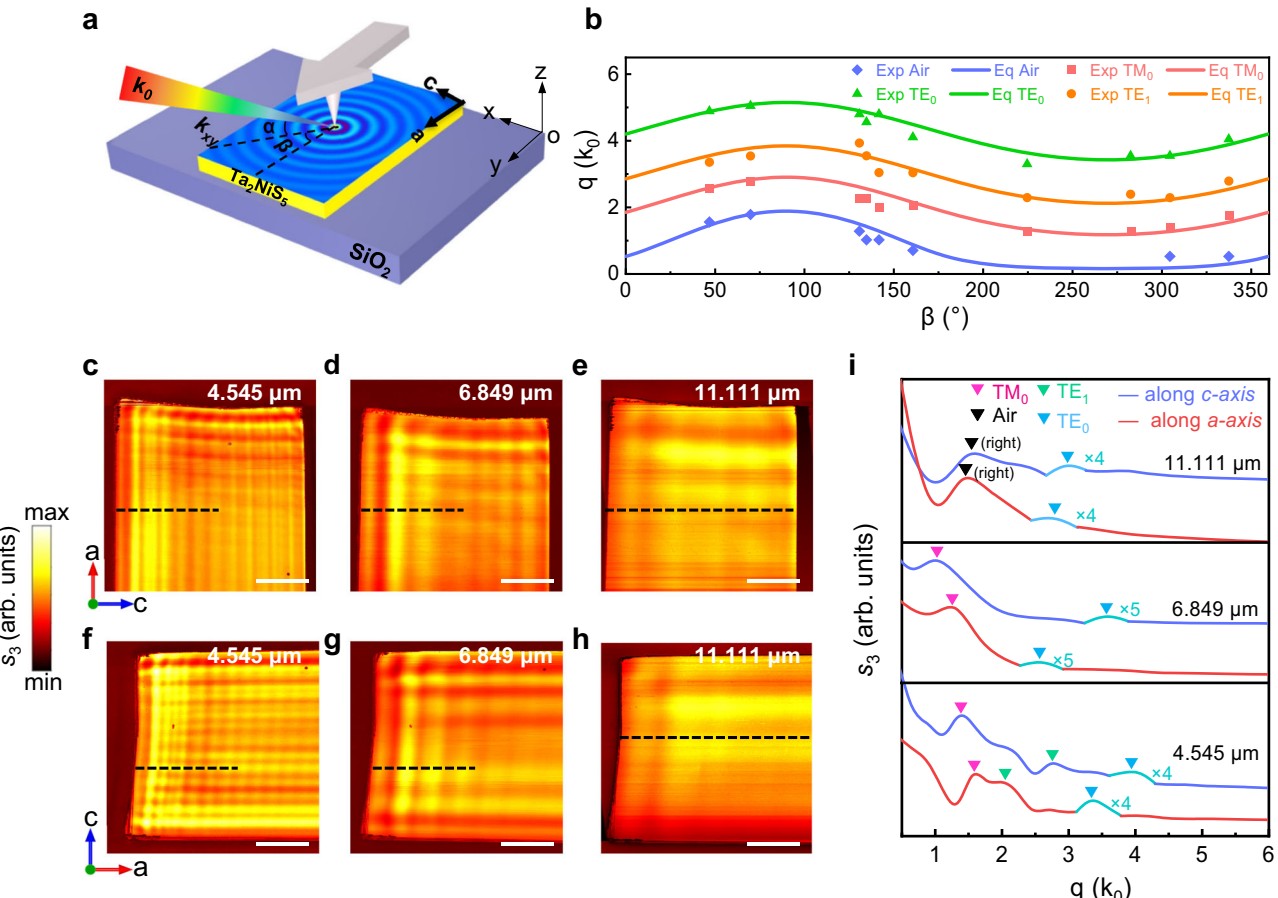

**Fig. 2 | Real-space nanoimaging of waveguide modes in Ta₂NiS₅ flakes. a** Three-dimensional schematic of the near-field experimental setup. $\alpha$ is the angle between the incident light wavevector ($k_0$) and sample surface (x–y plane), $\beta$ is the angle between the projection of the incident light in the x–y plane ($k_{xy}$) and the sample's edge ($a$-axis). **b** Experimental data points overlaid on the data calculated through theoretical analysis of an 881-nm-thick Ta₂NiS₅ sample on SiO₂ substrates under different rotating angles $\beta$. TE₀ and TE₁ represent zero-order and first-order transverse electric modes, TM₀ represents zero-order transverse magnetic modes and air represents air modes. **c–h** Near-field amplitude images of a 1002-nm-thick Ta₂NiS₅ flake under various excitation wavelengths, labeled in the upper right corner of each image. The sample edge on the left corresponds to $a$-axis of Ta₂NiS₅ flake in (**c–e**), while the edge on the left corresponds to $c$-axis in (**f–h**). scale bar: 10 μm. **i** Fourier transform (FT) analysis of (**c–h**). The fringe profiles of FT are extracted along the black dotted lines in (**c–h**) and different modes are marked by the colored inverted triangles. The label 'right' indicates the FT peak excited by the right edge.

where $\gamma = \sin^{-1}((\frac{k_0}{q_{wm}})\cos\alpha\cos\beta)$. Thereby, the wavevector $q_{wm}$ can be extracted from the complicated interference patterns.

To study the waveguide modes inside the Ta₂NiS₅ sample, we conducted a near-field research of the sample-edge-orientation dependence of the fringe patterns through the s-SNOM. The near-field images of Ta₂NiS₅ under different rotation angles $\beta$ are shown in Supplementary Figs. 10 and 11. Accordingly, we show that the fringe patterns vary systematically with $\beta$, and the theoretical calculation of each mode (solid line) is consistent with the experimental results (point) in Fig. 2b. This means that the interference fringes in Ta₂NiS₅ originate from the interference of dielectric waveguide modes and the incident light. Figure 2c–h show the oscillating fringes inside the Ta₂NiS₅ flake along two orthogonal crystallographic orientations when irradiated by various MIR excitation wavelengths. It can be clearly seen that the oscillating fringes occur inside all the sample flakes and show distinct behavior along two orthogonal crystallographic orientations (near-field amplitude images of another flake are shown in Supplementary Fig. 12). Moreover, the fringes spacing increases as the excitation wavelength is longer, which performs as the dispersion behavior of a dielectric waveguide mode.

In order to quantitatively analyze these complicated fringe profiles, we performed Fourier transform (FT) on the real-space fringe profiles. The results plotted in Fig. 2i show the FT intensities of the line profiles in Fig. 2c–h. The FT analysis data are extracted along the dash line in Fig. 2c–h. The results plotted show that the FT intensities are dominated by multiple peaks, whose types are verified by the calculated results with Eqs. (1), (2) and (3) (the refractive indices $n_b$ was set to a fixed value $2.45 + 0.02i$ to get a consistent result and the constant value is also agreed with the DFT calculation result). As shown in Fig. 2i, the difference in FT intensity along different axis is apparent, which suggests the existence of anisotropic waveguide mode inside Ta₂NiS₅ flakes. (Experimental dispersion data points and theoretical dispersion relations are shown in Supplementary Fig. 13).

## Anisotropic waveguide modes from visible to MIR

To further study the anisotropic waveguide modes in Ta₂NiS₅, we conducted a study on their dispersion relationships. Figure 3 shows the anisotropic waveguide dispersion relationships under visible excitation wavelengths. The near-field images and the corresponding fringe profiles (red lines) of an 83-nm-thick Ta₂NiS₅ with $a$-axis on the left and $c$-axis on the left under 633 nm and 785 nm excitation wavelength were plotted Figs. 3a, b, respectively. We conducted FT analysis of line profiles in Fig. 3a, b, and the corresponding waveguide modes points superimposed on the theoretical dispersion curves are shown in Fig. 3c, d. The theoretical dispersion curves of the waveguide modes were calculated by solving the transcendental Eqs. (1) and (2). Notably,

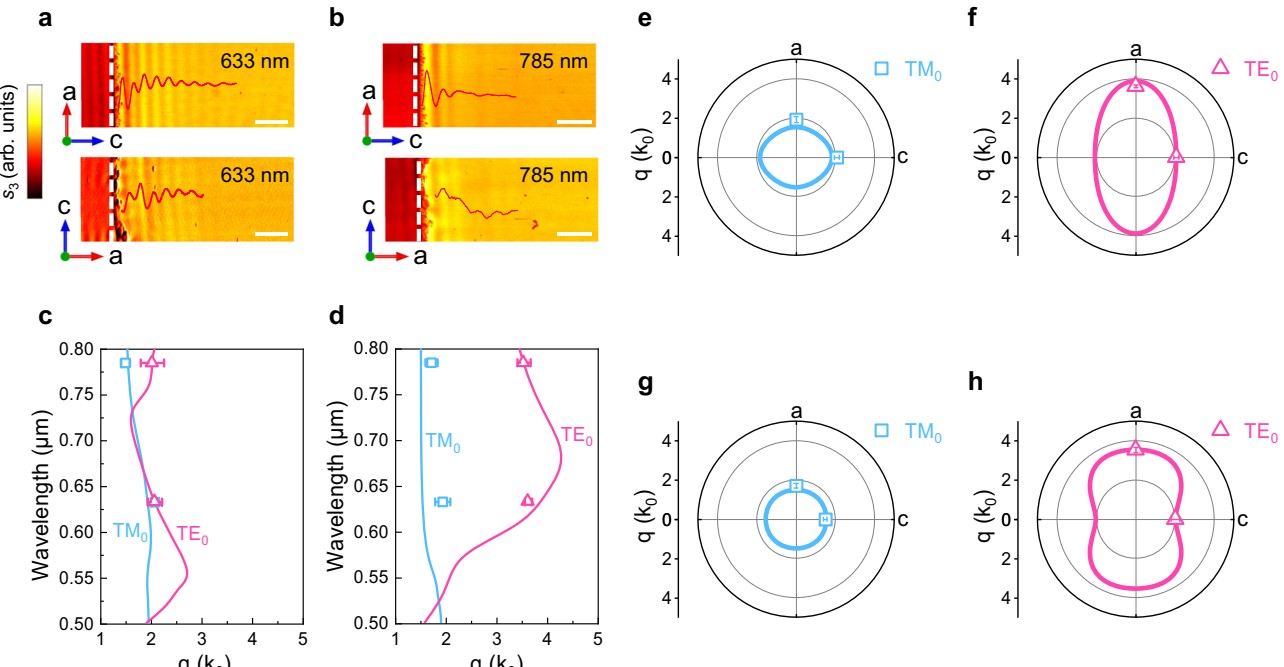

**Fig. 3 | In-plane anisotropic dispersion of the waveguide modes at visible range.** Near-field images of Ta$_2$NiS$_5$ flake with *a*-axis on the left and *c*-axis on the left under (**a**) 633 nm (**b**) 785 nm excitation wavelengths, labeled in the upper right corner of each image. The white dash line represents the edge of the sample. The corresponding fringe profiles of the images are represented by the red lines. Scale bar: 1 μm. **c**, **d** Experimental dispersion data (points) and theoretical dispersion relations of TM and TE polarized waveguide modes along *c*-axis and *a*-axis, respectively. Polar plots of wavevector *q* along different in-plane crystallographic directions of the TM and TE modes under excitation wavelengths of (**e**, **f**) 633 nm and (**g**, **h**) 785 nm. The error bars were determined from the half of the full-width at half-maximum (FWHM) of the peak in the Fourier transform (FT) analysis[69].

the dispersion of TE mode shows a larger difference along the two orthogonal directions *a/c* than that of TM mode. For clarity, we plot the wavevector of TM and TE modes in a polar coordinate under 633 and 785 nm excitation wavelength in Fig. 3e–h. It is explicitly to see the distinct isofrequency contours of TM and TE mode in Fig. 3e–h. At the excitation wavelength of 633 nm, the TM mode along the *a/c* direction has a ratio of $1.93k_O/2.07k_O$. Conversely, the TE mode along the *a/c* direction has a ratio of $3.62k_O/2.07k_O$. Meanwhile, we extracted the visible in-plane birefringence based on the near-field results, which is in good agreement with the spectroscopic ellipsometry results (Supplementary Note 8).

Figure 4 shows the dispersion relationships of anisotropic waveguide mode in a 1785-nm-thick Ta$_2$NiS$_5$ flake on CaF$_2$ substrate under MIR excitation wavelength (real-space fringe profiles and FT analysis see Supplementary Note 9). The dispersion relation of the waveguide modes was calculated via the imaginary part of the Fresnel reflectivity coefficients Im($r_s$) and Im($r_p$) with the transfer-matrix method[52–54] as shown in Fig. 4a–d. Figures 4a, b represent the dispersions of TM and TE modes along the *c*-axis respectively, and Fig. 4c, d represent the dispersions of TM and TE modes along the *a*-axis respectively. The different color branches corresponding to different order of TM and TE modes. As the wavevector of TM mode related with the out-of-plane refractive indices, a varied $n_b$ was used to fit the experiment data, which finally determined as a constant value of 2.45 + 0.02i and achieved a consistent result with experimental observations. To better support the extracted refractive index $n_b$, we provided error bars in the near-field experimental dispersion data points and calculated the error of the out-of-plane refractive index $n_b$. The extracted refractive index $n_b$ shows a small error within 10%, which indicates that the value of $n_b$ is reliable (Supplementary Note 10).

The discrete and clear dispersion features indicate low-loss waveguide modes propagating in the Ta$_2$NiS$_5$ flake. According to Fig. 2c–h, the fringes propagation over the image, which imply the

propagation length would probably over 20 μm at the MIR range. Such a propagation length is larger than the reported propagation length ~4 μm of dielectric waveguide mode found in the PtSe$_2$ at MIR range[52] (Supplementary Table 4) and is also consistent with the numerical results of 20 μm (*a*-axis) and 25 μm (*c*-axis) for TM mode, respectively (Supplementary Note 11). The thicknesses-dependence of dispersion relation in Ta$_2$NiS$_5$ is shown in Supplementary Fig. 17–19. The TE and TM wavevectors and modes numbers increase as the Ta$_2$NiS$_5$ flake thickness increases, and the trend is the same for samples on different substrates. Meanwhile, we plot the wavevector of TM and TE modes in a polar coordinate under the exciting wavelength of 4.545 and 6.849 μm in Fig. 4e–h. It is clearly to see a nearly circular shape of isofrequency contour of TM mode in Fig. 4e, g, otherwise TE mode exhibits an elliptical shape of isofrequency contour in Fig. 4f, h, which is consistent with the phenomenon under visible excitation wavelength. Hence, the TE waveguide modes in Ta$_2$NiS$_5$ show a significant potential to the directional control of light for nanophotonic applications.

## Discussion

Combining far- and near-field optical studies through the spectroscopic ellipsometry, FTIR and s-SNOM measurements, we have revealed that the layered ternary chalcogenides Ta$_2$NiS$_5$ presents both a giant in-plane and out-of-plane birefringence across visible to MIR wavelength ranges, which is highly beneficial for compact and integrated nanophotonic devices. Specifically, Ta$_2$NiS$_5$ exhibits superior in-plane birefringence and out-of-plane birefringence among vdW materials, showing a potential application for ultra-thin polarizer, waveplates, beam splitters, and phase-matching elements in MIR wavelengths. Through near-field nanoimaging, we demonstrate ultrawide-band incident light from visible to MIR region can be effectively localized and guided in the Ta$_2$NiS$_5$ flakes, and the dispersion relations of the waveguide modes are highly anisotropic. Such a

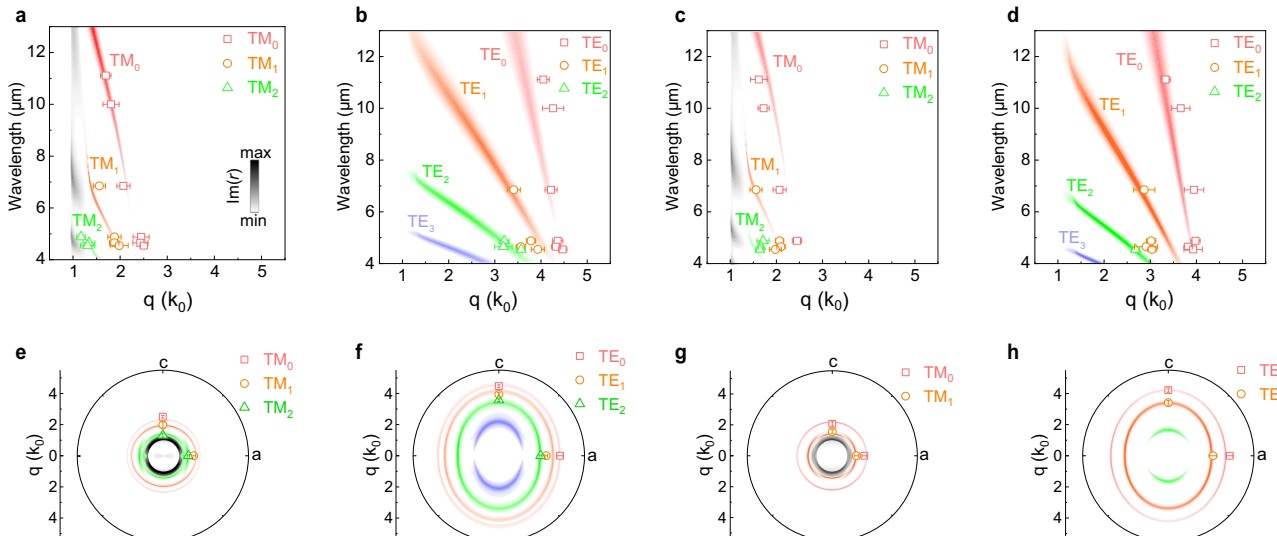

**Fig. 4 | In-plane anisotropic dispersion of the waveguide modes at MIR band.** Experimental dispersion data (points) and theoretical dispersion relations of TM and TE polarized waveguide modes along (**a**, **b**) $c$-axis and (**c**, **d**) $a$-axis, respectively. The $Ta_2NiS_5$ sample is on $CaF_2$ substrate and the thickness is 1785 nm. Im($r$) represents the imaginary part of the Fresnel reflectivity coefficients. The error bars were determined from the half of the full-width at half-maximum (FWHM) of the peak in the Fourier transform (FT) analysis[69]. Polar plot of wavevector $q$ along different in-plane crystallographic directions of the TM and TE modes under excitation wavelengths of (**e**, **f**) 4.545 μm and (**g**, **h**) 6.849 μm. The error bars were determined from the half of the FWHM of the peak in the FT analysis[69].

broadband, large refractive index, in-plane giant birefringence opens the door for lossless anisotropic dielectric waveguide modes transmission, which suggest great potential to directional control of light for on-chip compact nanophotonic applications. Moreover, we anticipate synthesizing vdW materials that support broadband giant birefringence and in-plane anisotropic waveguiding modes by changing the proportion or constituent of the elements in layered chalcogenides. For instance, the electronic polarizability of $Se^{2-}$ (10.5 Å$^3$) is slightly higher than $S^{2-}$ (10.2 Å$^3$)[15], and a large structural and polarizability anisotropy is expected to exist in ternary chalcogenides $Ta_2NiSe_5$ as well (Supplementary Table 1). We also found similar near-field propagating fringes in $Ta_2NiSe_5$ flakes (Supplementary Note 13). Our findings pave the way for utilizing layered biaxial chalcogenides as broadband giant birefringent material to develop subwavelength integrated optics in the future.

## Methods

### Sample preparation and characterization
Thin flakes of $Ta_2NiS_5$ were exfoliated from a bulk crystal (HQ graphene) onto $SiO_2$/Si (Si wafers with 290-nm-thick $SiO_2$ on top layer) or $CaF_2$ substrate using the mechanical exfoliation method.

### Spectroscopic ellipsometry
Spectroscopic ellipsometry measurements were performed with a commercial spectroscopic ellipsometer (J. A. Woollam, Inc. M2000X-FB-300XTF) at room temperature. The ellipsometric data Psi and Delta were collected from 300 to 800 nm at the incident angle of 65°. Considering the small lateral dimensions of the $Ta_2NiS_5$ flakes, they were measured by the focusing probes with a spot size of $60 \times 25$ μm$^2$ on the sample. The in-plane anisotropy of the $Ta_2NiS_5$ was measured by rotating the in-plane azimuth angles of the samples with a step of 10° from 0° to 360°.

### FTIR spectroscopy
MIR reflectance measurements were undertaken using a Bruker microscope coupled to a Bruker LUMOS II FTIR spectrometer (Bruker Optics GmbH, Ettlingen, Germany) equipped with a broadband MCT detector (600–4000 cm$^{-1}$). Off-normal (25° average incidence angle) polarized reflection were obtained from the crystals. The spectra were

collected with a 1 cm$^{-1}$ spectral resolution and the aperture size was set to $20 \times 20$ μm$^2$. All measurements were performed in reference to a gold film. All measurements were taken at room temperature and ambient pressure.

### Computational methods
All calculations are performed by using the Vienna Ab initio simulation package with the projector-augmented wave method[55]. The Perdew–Burke–Ernzerhof exchange–correlation functional and van der Waals corrections of DFT-D3 method are used for the geometry optimization and electronic properties calculations of $Ta_2NiS_5$[56,57]. GGA + U methods are applied to describe the strong correlation effects in the localized $d$ states with 5 eV for Ta atoms and 6 eV for Ni atoms, respectively[58]. The plane-wave cutoff energy is set as 500 eV and the separation of k-sampling for the first Brillouin zone is set as 0.025 Å$^{-1}$. The convergence of force and total energy is set as 0.01 eV Å$^{-1}$ and $10^{-6}$ eV, respectively.

### Near-field experiments
Near-field nanoimaging of the waveguide modes was performed with a commercial scattering-type scanning near-field optical microscope (NeaSNOM, NeaSpec GmbH). The samples were illuminated by monochromatic visible lasers of 633 or 785 nm and MIR quantum cascade lasers (QCL) (www.daylightsolutions.com) with tunable operating wavelengths from 4.545 to 11.111 μm (900–2200 cm$^{-1}$). The near-field images were obtained by pseudo-heterodyne interferometric detection module with tip-tapping frequency of about 270 kHz, the tip-tapping amplitudes are ~40 nm for the 633 and 785 nm wavelengths and ~80 nm for the MIR wavelengths. By demodulating the optical signal at the higher harmonic $n\Omega$ ($n \geq 3$), the noise from the background can be greatly suppressed, yielding near-field amplitude and phase images.

### Dispersion calculation
The imaginary part of the Fresnel reflectivity coefficients Im($r$) of the air/$Ta_2NiS_5$/$CaF_2$ multilayers (air/$Ta_2NiS_5$/$SiO_2$/Si in Supplementary Information) were calculated to study the anisotropic waveguide modes in $Ta_2NiS_5$, where the peaks of Im($r$) in the colormaps correspond to the maximum optical loss of incident light coupled to the

waveguide modes. The in-plane refractive indices $n_{a,c}$ of Ta$_2$NiS$_5$ was obtained from the far-field spectroscopic ellipsometry and FTIR measurements, and the out-of-plane refractive indices $n_b$ was set to 2.45 + 0.02i to fit a consistent result with experimental dispersion data extracted from the near-field experiments. The dielectrics of the CaF$_2$ and SiO$_2$/Si substrates were obtained from the previous literatures[59–61].

## Data availability

All technical details for producing the figures are enclosed in the supplementary information. All raw data generated during the current study are available from the corresponding authors upon request.

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

## Acknowledgements

We acknowledge professor Zhiming Shi from State Key Laboratory of Luminescence and Applications, Changchun Institute of Optics, Fine Mechanics and Physics for supporting DFT calculations. S.L. acknowledges the financial support from the National Natural Science Foundation of China (No. 62121005, 62022081, 61974099, 62075070 and 62334010), the National Key Research and Development Program (2021YFA0717600), the Natural Science Foundation of Jilin Province (20210101173JC) and Changchun Key Research and Development Program (21ZY03). R.Z. acknowledges the financial support from the NSAF (No. U2230108) and the National Key R&D Program of China (2021YFB2012601). R.C. acknowledges financial support from China Postdoctoral Science Foundation (2021M701298).

## Author contributions

Y.F., R.C. and J.H. contributed equally to this work. S.L., P.L., R.Z. and D.L. conceived of the original concept and supervised the project. S.L., Y.F., J.H., Y.Z., T.S., W.S., C.H. and X.S. performed the experiments and analysed the data. R.C., X.Z., W.L., W M. and L.Q. contributed to the theoretical analyses and modeling of the data. All authors discussed the results and co-wrote the paper. All authors have approved the final version of the manuscript.

## Competing interests

The authors declare no competing interests.
