## [Peer Review File · Nature Communications]

Visible to Mid-infrared Giant In-Plane Optical Anisotropy in Ternary Van der Waals CrystalsREVIEWER COMMENTS

Reviewer #1 (Remarks to the Author):

Feng and coauthors report the far-field and near-field study of optical anisotropy in Ta₂NiS₅ crystals. This manuscript shows the anisotropic responses of Ta₂NiS₅ in visible and mid infrared, and claims that they observed the highest birefringence. The research topic and the material are interesting. However, I do have some concerns and comments about the current manuscript.

1. Some similar works using near-field microscopy to demonstrate giant anisotropy have been published. For example, the work by Deng et al., *Nanotechnology*, 33, 345201 (2022), and several others. The birefringence was reported to be 2.

2. This manuscript reports in-plane and out-of-plane birefringence in the visible and mid infrared, respectively. I am not sure if these values are perfectly accurate. If I understand correctly, the out-of-plane birefringence was obtained by fitting the calculation with their near-field results with an adjustable parameter n_{-b} (out-of-plane). n_{-b} plays a vital role in determining the birefringence. Since about half of their data points do not have an excellent agreement with the calculation and no visible anisotropy data + theory are provided, see figure 3, this parameter extraction may be questionable. Error bars should also be provided to better support the extracted birefringence. In addition, since the authors do have near-field images in the visible in the supporting materials, it would be more robust if the visible in-plane birefringence could be extracted based on the near-field images. This may be particularly desirable since a high value of 2 was claimed in the manuscript as probably the key point. Results from the ellipsometry seem a bit thin to some degree.

3. Part of the manuscript is hard to follow. In the key part reporting their data, pages 9 to 11, there are lots of citations of the supplementary material figures, equations, and notes, which makes the current main text informationless. The authors may consider moving the visible near-field data, including the images and analysis, to the main text, since visible and mid infrared are equally emphasized in the manuscript. Also, there are no descriptions of dashed lines in figure 2.

4. In figure 3, why TE modes could not be observed in visible frequencies? In addition, data and theory on visible anisotropy may be provided in figure 3.

5. Some phrases in the manuscript may be reconsidered. In the discussion section of the manuscript, there are phrases claiming the propagation length is the highest, much higher than all previous works. I would suggest the authors reconsider these phrases since they also need to consider the confinement, which is pretty low in this work. Frankly, photons can propagate forever with low confinement. In addition, the authors claim their findings pave the way to develop ultraminiaturized integrated optics. Since the confinement is pretty low, how can the ultraminiaturization be achieved?

Reviewer #2 (Remarks to the Author):

The authors report on large broadband optical anisotropy in Ta₂NiS₅. Recently, there is a lot of interest in discovering crystals with large optical anisotropy. Typically, van der Waals materials have large out of plane anisotropy but the in plane anisotropy remains modest. This work reports large in plane anisotropy in a ternary van der Waals crystals, Ta₂NiS₅ and hints at potentially large anisotropy in Ta₂NiSe₅. I have a few comments and criticisms on the manuscript.

1. What are the absorption edges along a-, b- and c- axes? It seems like along a- and c-, it is much longer than 800 nm (Figure 1d). In this case, it is meaningless to talk about large visible birefringence. Birefringence is only meaningful in fully transparent regions for two different axes. Even if one were to relax this, the current materials seems to absorb in at least two the three directions.
2. Please note a recent preprint on large optical anisotropy in Sr_{1+x}TiS₃ (<https://arxiv.org/abs/2303.00041>) from similar group of researchers that performed the study on BaTiS₃. Perhaps, it is worth discussing the current results in the light of this information.
3. Is there a reason to cut off the data presented between 2.5 - 4 micron in the FTIR measurements?
4. As the authors have data on the dichroism (Δk), it may be worth including that in the manuscript and briefly discussing it.
5. Did the authors perform DFT calculations to see if they can reproduce the experimental values in the theoretical models? Although polarizability of the species is a good argument, it is unclear whether that is a sufficiently rigorous argument.
6. What is the origin of the uptick in the k for a- and c- axes > 10 micron? Is it free carriers or phonons? It will be useful to know if there is any reported information on their electrical characteristics. Perhaps it can also help understand the SNOM results better.

Overall this is an interesting manuscript, but some of the open issues noted above have to be addressed before I can recommend this manuscript for publication.

Manuscript ID: NCOMMS-23-27157

Title: Visible to Mid-infrared Giant In-Plane Optical Anisotropy in Ternary Van der Waals Crystals

Point-by-point responses to Reviewers' Comments

We sincerely thank the reviewers for the insightful comments concerning our manuscript. These comments are very valuable for further improving the quality of this manuscript. In the revised manuscript, we have tried our best to address all the concerns raised by the reviewers, and the point-by-point responses are listed below. The changes to the manuscript are indicated in **RED** in the marked version of the revised manuscript. We hope that the quality of our manuscript has met the publication standard after this revision.

Reviewer #1 (Remarks to the Author):

Feng and coauthors report the far-field and near-field study of optical anisotropy in Ta₂NiS₅ crystals. This manuscript shows the anisotropic responses of Ta₂NiS₅ in visible and mid infrared, and claims that they observed the highest birefringence. The research topic and the material are interesting. However, I do have some concerns and comments about the current manuscript.

[Reply]

We thank the reviewer for his/her positive comments. We have tried our best to address the concerns raised by the reviewer, as listed below.

[Comment 1] *Some similar works using near-field microscopy to demonstrate giant anisotropy have been published. For example, the work by Deng et al., Nanotechnology, 33, 345201 (2022), and several others. The birefringence was reported to be 2.*

[Reply]

Thanks for the comment. We have added a discussion about the above works in the "Introduction" section of the revised manuscript and the paper mentioned by the

reviewer have been cited as ref. [22] in the revised manuscript. The work by Deng et al. reported the out-of-plane birefringence of WS₂, the optic axis of WS₂ is out of the plane, and thus its utility in conventional optical systems is limited by the difficulty to funnel light into the small nanoscale area that parallels to its out-of-plane optical axis. In our work, we mainly focus on giant in-plane birefringence of the vdW crystal Ta₂NiS₅. Compared with out-of-plane birefringence, the realization of large in-plane birefringence is more challenging and more conducive to ultracompact integrated applications.

[Actions]

On page 2-3 of the revised manuscript, we added the above paper as ref. [22], and changed the sentence to “The latest researches have suggested layered van der Waals (vdW) materials exhibit inherent large birefringence, as the weak interlayer bonding leads naturally to considerable out-of-plane birefringence (eg. MoS₂^{20,21} (~3), WS₂ (~2)²² and h-BN²³ (~0.7)).”

[Comment 2a] *This manuscript reports in-plane and out-of-plane birefringence in the visible and mid infrared, respectively. I am not sure if these values are perfectly accurate. If I understand correctly, the out-of-plane birefringence was obtained by fitting the calculation with their near-field results with an adjustable parameter n_b (out-of-plane). n_b plays a vital role in determining the birefringence. Since about half of their data points do not have an excellent agreement with the calculation and no visible anisotropy data theory are provided, see figure 3, this parameter extraction may be questionable. Error bars should also be provided to better support the extracted birefringence.*

[Reply]

Thank the reviewer for raising this good point. We would like to elaborate on this point as below.

First, using the near-field imaging to extract the out-of-plane refractive index has been proved to be an effective method [1] Hu, D. et al. *Probing optical anisotropy of*

nanometer-thin van der waals microcrystals by near-field imaging. Nat. Commun. 8, 1471 (2017);

2) Ermolaev, G.A. et al. *Giant optical anisotropy in transition metal dichalcogenides for next-generation photonics*. Nat. Commun. 12, 854 (2021)], we utilized this method to extract the refractive index n_b in our experiments. According to the transcendental equations R1 as shown below (equation (2) in the main text), the error of n_b mainly origins from the uncertainty of determining the wave vector q_{wm} from the near-field experimental data.

$$\sqrt{\frac{n_a^2}{n_b^2}(n_b^2 k_0^2 - q_{wm}^2)} d = \tan^{-1} \left(\frac{n_a^2 \sqrt{q_{wm}^2 - k_0^2 n_1^2}}{n_1^2 \sqrt{\frac{n_a^2}{n_b^2}(n_b^2 k_0^2 - q_{wm}^2)}} \right) + \tan^{-1} \left(\frac{n_a^2 \sqrt{q_{wm}^2 - k_0^2 n_s^2}}{n_s^2 \sqrt{\frac{n_a^2}{n_b^2}(n_b^2 k_0^2 - q_{wm}^2)}} \right) + n\pi \quad (\text{R1})$$

Thereby, to better support the extracted refractive index, we provided error bars in the near-field experimental dispersion data points (see below revised Figure **R1**), in which the error bars are determined by the half of the full-width at half-maximum (FWHM) of the peak in the Fourier transform (FT) analysis. [1] Babicheva, V. E. et al. *Near-Field Surface Waves in Few-Layer MoS₂*. ACS Photonics 5, 2106-2112 (2018); 2) Gesuele, F. et al. *Towards routine near-field optical characterization of silicon-based photonic structures: An optical mode analysis in integrated waveguides by transmission AFM-based SNOM*. Physica E: Low-dimensional Systems and Nanostructures 41, 1130-1134 (2009)]. By solving the above equation R1, we then obtain the error of the out-of-plane refractive index n_b , as shown in below **Table I**. Notably, the extracted refractive index n_b shows a small error within 10%, which indicates that the value of n_b is reliable.

Figure R1. In-plane anisotropic dispersion of the waveguide modes at MIR band. Experimental

dispersion data (points) and theoretical dispersion relations of TM and TE polarized waveguide modes along **(a, b)** *c*-axis and **(c, d)** *a*-axis, respectively. The Ta₂NiS₅ sample is on CaF₂ substrate and the thickness is 1785 nm. **e-h** Polar plot of wavevector *k* along different in-plane crystallographic directions of the TM and TE modes under excitation wavelengths of **(e, f)** 4.545 μm and **(g, h)** 6.849 μm. The error bars were determined from the half of the FWHM of the peak in the FT analysis.

Table I. The extract n_b value based on the near-field experimental data

Wavelength	Near-field data points	n_b
4.545 μm	TM ₀	2.63±0.08
	TM ₁	2.47±0.25
4.651 μm	TM ₀	2.56±0.17
	TM ₁	2.36±0.13
4.878 μm	TM ₀	2.58±0.18
	TM ₁	2.45±0.19
6.849 μm	TM ₀	2.33±0.16
	TM ₁	3.08±0.54
10 μm	TM ₀	2.36±0.25
11.111 μm	TM ₀	2.40±0.19

Second, we also noticed that the main mismatch between the experimental and calculation dispersion data points occur at 10 μm and 11.111 μm in Figure 3 of our previous submission. After further discussion, we found that since few fringes are excited under 10 and 11.111 μm compared to the case of shorter wavelength excitation, a wider range should be measured in the experiment so as to obtain a more accurate FT analysis. Thereby, we measured the near-field fringe of the same crystal across a wider range and re-extracted the experimental data at 10 and 11.111 μm. The results are shown in below revised Figure R2. We have also updated these data points in above Figure R1. Notably, the extracted modes are in good agreement with our theoretical simulation, as shown in Figure R1.

Figure R2. **a** Near-field image of a 1785-nm-thick Ta₂NiS₅ flake on CaF₂ substrate with *a*-axis on the left under 10 μm and 11.111 μm excitation wavelengths, labeled in the upper right corner of each image. Scale bar: 4 μm. **b, c** Real-space fringe profiles and the corresponding FT profiles of **a**. **d** Near-field image of a 1785-nm-thick Ta₂NiS₅ flake on CaF₂ substrate with *c*-axis on the left under 10 μm and 11.111 μm excitation wavelengths, labeled in the lower right corner of each image. Scale bar: 4 μm. **e, f** Real-space fringe profiles and the corresponding FT profiles of **d**. The label ‘(right)’ in **c** and **f** indicates the FT peak excited by the right edge.

Third, we have also followed the reviewer’s suggestion and provided visible anisotropy data in the revised Figure 3 of the main text (see below Figure 3). We have extracted the visible in-plane birefringence based on the near-field data points using transcendental equation **R2**.

$$\sqrt{n_c^2 k_0^2 - q_{wm}^2} d = \tan^{-1} \left(\frac{\sqrt{q_{wm}^2 - k_0^2 n_1^2}}{\sqrt{n_c^2 k_0^2 - q_{wm}^2}} \right) + \tan^{-1} \left(\frac{\sqrt{q_{wm}^2 - k_0^2 n_s^2}}{\sqrt{n_c^2 k_0^2 - q_{wm}^2}} \right) + m\pi \quad (\text{R2})$$

According to above equation, the in-plane birefringence of the materials is 2.04 under 633 nm excitation wavelength and 0.64 under 785 nm excitation wavelength, which is in good agreement with the ellipsometry results (see below Figure **R3**). Furthermore, the extracted n_b according to equation **R1** is around 2.8 ± 0.2 , which is generally in agreement with the value used in the main text.

Figure R3. The extracted in-plane birefringence data from near-field imaging (blue points) and the spectroscopic ellipsometry (red line) of Ta_2NiS_5 .

Figure 3. In-plane anisotropic dispersion of the waveguide modes at visible range. Near-field images of Ta_2NiS_5 flake with a -axis on the left and c -axis on the left under **a** 633nm **b** 785nm excitation wavelengths, labeled in the upper right corner of each image. The white dash line represents the edge of the sample. The corresponding fringe profiles of the images are represented by the red lines. **c, d** Experimental dispersion data (points) and theoretical dispersion relations of TM and TE polarized waveguide modes along c -axis and a -axis, respectively. **e-h** Polar plots of wavevector k along different in-plane crystallographic directions of the TM and TE modes under excitation wavelengths of (**e, f**) 633 nm and (**g, h**) 785 nm. The error bars were determined from the half of the FWHM of the peak in the FT analysis.

[Actions]

On page 12 of the revised main text, we have added error bars in the near-field experimental dispersion data. The updated figure has been shown as Figure 4 in the revised version of the manuscript.

Figure 4. In-plane anisotropic dispersion of the waveguide modes at MIR band. Experimental dispersion data (points) and theoretical dispersion relations of TM and TE polarized waveguide modes along **(a, b)** *c*-axis and **(c, d)** *a*-axis, respectively. The Ta₂NiS₅ sample is on CaF₂ substrate and the thickness is 1785 nm. **e-h** Polar plot of wavevector *k* along different in-plane crystallographic directions of the TM and TE modes under excitation wavelengths of **(e, f)** 4.545 μm and **(g, h)** 6.849 μm. The error bars were determined from the half of the FWHM of the peak in the FT analysis.

On page 10 of the revised main text, we have provided visible anisotropy data and added error bars as well. The figure has been shown as Figure 3 in the revised version of the manuscript.

Figure 3. In-plane anisotropic dispersion of the waveguide modes at visible range. Near-field images of Ta₂NiS₅ flake with *a*-axis on the left and *c*-axis on the left under **a** 633nm **b** 785nm excitation wavelengths, labeled in the upper right corner of each image. The white dash line represents the edge of the sample. The corresponding fringe profiles of the images are represented by the red lines. **c, d** Experimental dispersion data (points) and theoretical dispersion relations of TM and TE polarized waveguide modes along *c*-axis and *a*-axis, respectively. **e-h** Polar plots of wavevector *k* along different in-plane crystallographic directions of the TM and TE modes under excitation wavelengths of (**e, f**) 633 nm and (**g, h**) 785 nm. The error bars were determined from the half of the full-width at half-maximum (FWHM) of the peak in the FT analysis⁵⁸.

On page 17-18 of the revised SI, we also added a few sentences about the extracted n_b error from the near-field results as **Supplementary Note 10**.

According to the transcendental equations (2) as shown in the main text, the error of n_b mainly originates from the uncertainty of determining the wave vector q_{wm} from the near-field experimental data. To better support the extracted refractive index, we provided error bars in the near-field experimental dispersion data (**Figure 3 and Figure 4** of the main text), in which the error bars are determined by the half of the full-width at half-maximum (FWHM) of the peak in the Fourier transform (FT) analysis²⁰. By solving the transcendental equations, we then obtain the error of the out-of-plane refractive index n_b , as shown in **Supplementary Table 2**. Notably, the extracted refractive index n_b shows a small error within 10%, which indicates that the value of n_b

is reliable.

Supplementary Table 2. The extracted n_b value based on the near-field experimental data

Wavelength	Near-field data points	n_b
4.545 μm	TM ₀	2.63 \pm 0.08
	TM ₁	2.47 \pm 0.25
4.651 μm	TM ₀	2.56 \pm 0.17
	TM ₁	2.36 \pm 0.13
4.878 μm	TM ₀	2.58 \pm 0.18
	TM ₁	2.45 \pm 0.19
6.849 μm	TM ₀	2.33 \pm 0.16
	TM ₁	3.08 \pm 0.54
10 μm	TM ₀	2.36 \pm 0.25
11.111 μm	TM ₀	2.40 \pm 0.19

On page 16 of the revised SI, we have also updated the near-field image and FT analysis under 10 μm and 11.111 μm excitation.

Supplementary Figure 15. Near-field nanoimaging of the in-plane anisotropic waveguide modes at MIR band. **a, b** Optical microscope images of a 1785-nm-thick Ta₂NiS₅ flake on CaF₂

substrate with **a** *a*-axis on the left **b** *c*-axis on the left. Scale bar: 20 μm . **c, d** Near-field image along *c*-axis (**S15a** blue dash line) and *a*-axis (**S15b** red dash line), respectively. Scale bar: 4 μm . **e, f** Real-space fringe profiles and the corresponding FT profiles of **c**. **g, h** Real-space fringe profiles and the corresponding FT profiles of **d**. The label '(right)' in **f** and **h** indicates the FT peak excited by the right edge.

On page 12 of the revised main text, we have added the sentences “To better support the extracted refractive index n_b , we provided error bars in the near-field experimental dispersion data points and calculated the error of the out-of-plane refractive index n_b . The extracted refractive index n_b shows a small error within 10%, which indicates that the value of n_b is reliable (**Supplementary Note 10**).”

[Comment 2b] *In addition, since the authors do have near-field images in the visible in the supporting materials, it would be more robust if the visible in-plane birefringence could be extracted based on the near-field images. This may be particularly desirable since a high value of 2 was claimed in the manuscript as probably the key point. Results from the ellipsometry seem a bit thin to some degree.*

[Reply]

Thank the reviewer for the constructive suggestion. We have followed the reviewer’s suggestion and extracted the visible in-plane birefringence based on the near-field results using below transcendental equation.

$$\sqrt{n_c^2 k_0^2 - q_{wm}^2} d = \tan^{-1} \left(\frac{\sqrt{q_{wm}^2 - k_0^2 n_1^2}}{\sqrt{n_c^2 k_0^2 - q_{wm}^2}} \right) + \tan^{-1} \left(\frac{\sqrt{q_{wm}^2 - k_0^2 n_s^2}}{\sqrt{n_c^2 k_0^2 - q_{wm}^2}} \right) + m\pi$$

According to above equation, the in-plane birefringence of the materials is 2.04 under 633 nm excitation wavelength and 0.64 under 785 nm excitation wavelength, which is in good agreement with the spectroscopic ellipsometry results (see Figure **R3**).

Figure R3. The extracted in-plane birefringence data from near-field imaging (blue points) and the spectroscopic ellipsometry (red line) of Ta_2NiS_5 .

[Actions]

On page 15-16 of the revised SI, we added a paragraph on the extraction of visible in-plane birefringence from near-field imaging as **Supplementary Note 8**.

“To further confirm the in-plane birefringence of Ta_2NiS_5 , we extracted the visible in-plane birefringence based on the near-field results using transcendental equation (1) as mentioned in the main text. The birefringence of the materials is 2.04 under 633 nm excitation wavelength and 0.64 under 785 nm excitation wavelength, which is in good agreement with the spectroscopic ellipsometry results (Supplementary Figure 14).”

Supplementary Figure 14. The extracted in-plane birefringence data from near-field imaging (blue points) and the spectroscopic ellipsometry (red line) of Ta_2NiS_5 .

On page 11 of the revised main text, we added a sentence “Meanwhile, we extracted

the visible in-plane birefringence based on the near-field results, which is in good agreement with the spectroscopic ellipsometry results (**Supplementary Note 8**).”

[Comment 3] *Part of the manuscript is hard to follow. In the key part reporting their data, pages 9 to 11, there are lots of citations of the supplementary material figures, equations, and notes, which makes the current main text informationless. The authors may consider moving the visible near-field data, including the images and analysis, to the main text, since visible and mid infrared are equally emphasized in the manuscript. Also, there are no descriptions of dashed lines in figure 2.*

[Reply]

Thank the reviewer for the constructive suggestion. We have followed the reviewer’s suggestion and rewrote the main text in pages 9 to 11 to make our manuscript clear and concise. We moved the visible near-field data, including the images and analysis, to the main text (as new revised **Figure 3**) in the revised manuscript.

We also added a description of the dashed line in Figure 2 ' The FT analysis data are extracted along the dash line in **Figure 2c-h.**' in the revised main text.

[Actions]

On page 11 of the revised main text, we have added a new paragraph about the visible near-field results shown in the revised **Figure 3**.

“To further study the anisotropic waveguide modes in Ta₂NiS₅, we conducted a study on their dispersion relationships. **Figure 3** shows the anisotropic waveguide dispersion relationships under visible excitation wavelengths. The near-field images and the corresponding fringe profiles (red lines) of an 83-nm-thick Ta₂NiS₅ with *a*-axis on the left and *c*-axis on the left under 633 nm and 785 nm excitation wavelength were plotted **Figure 3a** and **3b**, respectively. We conducted FT analysis of line profiles in **Figure 3a** and **3b**, and the corresponding waveguide modes points superimposed on the theoretical dispersion curves are shown in **Figure 3c** and **3d**. The theoretical dispersion curves of the waveguide modes were calculated by solving the transcendental equations (1) and (2). Notably, the dispersion of TE mode shows a larger difference along the two

orthogonal directions a/c than that of TM mode. For clarity, we plot the wavevector of TM and TE modes in a polar coordinate under 633 and 785 nm excitation wavelength in **Figure 3e-h**. It is explicitly to see the distinct isofrequency contours of TM and TE mode in **Figure 3e-3h**. At the excitation wavelength of 633 nm, the TM mode along the a/c direction has a ratio of $1.93k_0/2.07k_0$. Conversely, the TE mode along the a/c direction has a ratio of $3.62k_0/2.07k_0$. Meanwhile, we extracted the visible in-plane birefringence based on the near-field results, which is in good agreement with the spectroscopic ellipsometry results (**Supplementary Note 8**).

Figure 3. In-plane anisotropic dispersion of the waveguide modes at visible range. Near-field images of Ta₂NiS₅ flake with a -axis on the left and c -axis on the left under **a** 633nm **b** 785nm excitation wavelengths, labeled in the upper right corner of each image. The white dash line represents the edge of the sample. The corresponding fringe profiles of the images are represented by the red lines. **c, d** Experimental dispersion data (points) and theoretical dispersion relations of TM and TE polarized waveguide modes along c -axis and a -axis, respectively. **e-h** Polar plots of wavevector k along different in-plane crystallographic directions of the TM and TE modes under excitation wavelengths of (**e, f**) 633 nm and (**g, h**) 785 nm. The error bars were determined from the half of the full-width at half-maximum (FWHM) of the peak in the FT analysis⁵⁸.

On page 10 of the revised main text, we have added a description of the dashed line in Figure 2 ' The FT analysis data are extracted along the dash line in **Figure 2c-h**. ' in the revised main text.

[Comment 4] *In figure 3, why TE modes could not be observed in visible frequencies? In addition, data and theory on visible anisotropy may be provided in figure 3.*

[Reply]

Thanks for raising this point. We would like to explain this point based on the following analyses.

First, in the visible regime, the number of waveguide modes increases with the thickness of the Ta₂NiS₅ layer. The plane wave can excite a set of modes that related to the incident angle [Hermann, R. et al. *Mode launching on a multimode slab-waveguide by a plane wave*. Appl. Phys. 9, 307-313 (1976)]. Meanwhile, the extracted experimental wave vector in thicker sample fall in the higher order modes [Hu, D. et al. *Probing optical anisotropy of nanometer-thin van der waals microcrystals by near-field imaging*. Nat. Commun. 8, 1471 (2017)], and the peak of FT analysis may be a mixture of multiple modes. Therefore, For the thick sample (1785 nm) demonstrated in the **Figure 3** of our previous submission, it is difficult to identify and distinguish the corresponding modes (Figure **R4**).

Second, to better illustrate the corresponding modes in visible frequencies, we have analysed a thin (83-nm-thick) sample, as illustrated in below Figure **R5**. It can be observed that for a thinner sample of 83 nm in thickness, only fundamental TM₀ and TE₀ mode are supported, and the experimental data are well consistent with the calculated results. We have also added the relevant data and theory on visible anisotropy as revised **Figure 3** of the manuscript.

Figure R4. **a** Near-field images of a 1785-nm-thick Ta_2NiS_5 flake on CaF_2 substrate with a -axis on the left under 633nm and 785nm excitation wavelengths, labeled in the upper right corner of each image. The white dash line represents the edge of the sample. Scale bar: 4 μm . **b** FT analysis of real-space fringe profiles in **a**. **c**, **d** Theoretical dispersion relations of **c** TM and **d** TE polarized waveguide modes along c -axis, respectively. **e** Near-field images of a 1785-nm-thick Ta_2NiS_5 flake on CaF_2 substrate with c -axis on the left under 633nm and 785nm excitation wavelengths, labeled in the upper right corner of each image. The white dash line represents the edge of the sample. Scale bar: 4 μm . **f** FT analysis of real-space fringe profiles in **e**. **g**, **h** Theoretical dispersion relations of **g** TM and **h** TE polarized waveguide modes along a -axis, respectively.

Figure R5. In-plane anisotropic dispersion of the waveguide modes at visible range. Near-field images of Ta₂NiS₅ flake with *a*-axis on the left and *c*-axis on the left under **a** 633nm **b** 785nm excitation wavelengths, labeled in the upper right corner of each image. The white dash line represents the edge of the sample. The corresponding fringe profiles of the images are represented by the red lines. **c, d** Experimental dispersion data (points) and theoretical dispersion relations of TM and TE polarized waveguide modes along *c*-axis and *a*-axis, respectively. **e-h** Polar plots of wavevector *k* along different in-plane crystallographic directions of the TM and TE modes under excitation wavelengths of (**e, f**) 633 nm and (**g, h**) 785 nm. The error bars were determined from the half of the FWHM of the peak in the FT analysis.

[Actions]

On page 19 of the revised SI, we added the analysis for the thicknesses-dependence of dispersion relationship under visible frequencies as **Supplementary Note 12** and updated the **Supplementary Figure 17**.

In the visible regime, the number of waveguide modes increases with the thickness of the Ta₂NiS₅ layer. For a waveguide with a 1002-nm-thick Ta₂NiS₅ sample, there are 6 TE modes and 13 TM modes in theory. Meanwhile, the extracted experimental wave vector in thicker sample falls in the higher order modes^{21,22}, and the peak of Fourier transform analysis may be a mixture of multiple modes. Therefore, it is difficult to identify the corresponding modes by the near-field data for the thick sample. On the contrary, for a thinner sample of 83 nm in thickness, only fundamental TM₀ and TE₀ mode are supported and the experimental data are well consistent with the calculated results.

Supplementary Figure 17. Near-field characteristics of Ta₂NiS₅ flakes with various thicknesses on SiO₂ substrate. a Near-field images of Ta₂NiS₅ flake with various thicknesses. The excitation wavelength $\lambda=633$ nm. The Ta₂NiS₅ flakes are placed *a*-axis on the left and fringes are extracted along *c*-axis. Scale bar: 1 μm . **b, c** Real-space fringe profiles and the corresponding FT profiles of **a**. **d, e** Experimental dispersion data points and theoretical dispersion relations of the TM- and TE-polarized waveguide modes of **a**.

On page 10 of the revised main text, we have provided visible anisotropy data and added error bars. The figure has been shown as Figure 3 in the revised version of the manuscript.

Figure 3. In-plane anisotropic dispersion of the waveguide modes at visible range. Near-field

images of Ta₂NiS₅ flake with *a*-axis on the left and *c*-axis on the left under **a** 633nm **b** 785nm excitation wavelengths, labeled in the upper right corner of each image. The white dash line represents the edge of the sample. The corresponding fringe profiles of the images are represented by the red lines. **c, d** Experimental dispersion data (points) and theoretical dispersion relations of TM and TE polarized waveguide modes along *c*-axis and *a*-axis, respectively. **e-h** Polar plots of wavevector *k* along different in-plane crystallographic directions of the TM and TE modes under excitation wavelengths of (**e, f**) 633 nm and (**g, h**) 785 nm. The error bars were determined from the half of the full-width at half-maximum (FWHM) of the peak in the FT analysis⁵⁸.

[Comment 5a] *Some phrases in the manuscript may be reconsidered. In the discussion section of the manuscript, there are phrases claiming the propagation length is the highest, much higher than all previous works. I would suggest the authors reconsider these phrases since they also need to consider the confinement, which is pretty low in this work. Frankly, photons can propagate forever with low confinement.*

[Reply]

Thanks for pointing out this inappropriate expression. We have revised and deleted the terms 'maximum' in the revised version.

[Actions]

On page 13 of the revised manuscript, we changed the sentence to “**Such a propagation length is larger than the reported propagation length ~ 4 μm of dielectric waveguide mode found in the PtSe₂ at MIR range⁵⁷ (Supplementary Table 3) and is also consistent with the numerical results of 20 μm for TE mode and 25 μm for TM mode, respectively (Supplementary Note 11).**”

[Comment 5b] *In addition, the authors claim their findings pave the way to develop ultraminiaturized integrated optics. Since the confinement is pretty low, how can the ultraminiaturization be achieved?*

[Reply]

Thanks for pointing out this inappropriate expression. We have changed the terms 'ultraminiaturized' to 'subwavelength' in the revised version.

[Actions]

On page 14 of the revised manuscript, we changed the sentence to “Our findings pave the way for utilizing layered biaxial chalcogenides as broadband giant birefringent material to develop subwavelength integrated optics in the future.”

Reviewer #2 (Remarks to the Author):

The authors report on large broadband optical anisotropy in Ta₂NiS₅. Recently, there is a lot of interest in discovering crystals with large optical anisotropy. Typically, van der Waals materials have large out of plane anisotropy but the in plane anisotropy remains modest. This work reports large in plane anisotropy in a ternary van der Waals crystals, Ta₂NiS₅ and hints at potentially large anisotropy in Ta₂NiSe₅. I have a few comments and criticisms on the manuscript.

Overall this is an interesting manuscript, but some of the open issues noted above have to be addressed before I can recommend this manuscript for publication.

[Reply]

We thank the reviewer for his/her positive comments. We have tried our best to address the concerns raised by the reviewer, as listed below.

[Comment 1] *What are the absorption edges along a-, b- and c- axes? It seems like along a- and c-, it is much longer than 800 nm (Figure 1d). In this case, it is meaningless to talk about large visible birefringence. Birefringence is only meaningful in fully transparent regions for two different axes. Even if one were to relax this, the current materials seems to absorb in at least two the three directions.*

[Reply]

Thanks for raising this good point. The absorption edges of Ta₂NiS₅ along a-, b- and c- axes have been investigated previously, and the values are in the range of 0.2-0.3 eV. [1) Li, L. et al. *Strong In-Plane Anisotropies of Optical and Electrical Response in Layered Dimetal Chalcogenide*. ACS Nano 11, 10264-10272 (2017); 2) Larkin, T. I. et al. *Giant exciton Fano resonance in quasi-one-dimensional Ta₂NiSe₅*. Phys. Rev. B 95, 195144 (2017).] And the interband transition-induced absorption occur along a- and c- axes around 800 nm and beyond [Larkin, T. I. et al. *Giant exciton Fano resonance in quasi-one-dimensional Ta₂NiSe₅*. Phys. Rev. B 95, 195144 (2017)]. We do agree with the reviewer that birefringence is meaningful in fully transparent regions for bulk birefringent materials, such as calcite and rutile. However, van der Waals materials are distinguished from the traditional bulk

materials owing to their very thin thickness. Light within their absorption range can still transmit through these materials after a short transmission distance, thereby enabling their potential application of birefringent properties. At present, the birefringence of van der Waals materials (MoS₂, FePS₃, etc.) has aroused increasing interests and proved to be beneficial for optical components including polarizers, waveplates, etc. [1) Ermolaev, G.A. et al. *Giant optical anisotropy in transition metal dichalcogenides for next-generation photonics*. Nat. Commun. **12**, 854 (2021); 2) Hu, D. et al. *Probing optical anisotropy of nanometer-thin van der waals microcrystals by near-field imaging*. Nat. Commun. **8**, 1471 (2017); 3) Yang, H. et al. *Optical waveplates based on birefringence of anisotropic two-dimensional layered materials*. ACS Photonics **4**, 3023-3030 (2017).] With this point of view, we discussed the birefringence as one of the fundamental optical properties of the Ta₂NiS₅ in our work, and hope that it will help in gaining a more comprehensive understanding of the optical properties of this novel and appealing material.

[Actions]

On page 5-6 of the revised main text, we added the sentence “**Given the very thin nature of vdW materials, light within their absorption range can still transmit through these materials after a short transmission distance, thereby enabling their potential application of birefringent properties.**”

[Comment 2] *Please note a recent preprint on large optical anisotropy in Sr_{1+x}TiS₃ (<https://arxiv.org/abs/2303.00041>) from similar group of researchers that performed the study on BaTiS₃. Perhaps, it is worth discussing the current results in the light of this information.*

[Reply]

Thanks for providing this useful information. The mentioned work reported on Sr_{9/8}TiS₃ as a bulk crystal material, which is basically different from the thin van der Waals crystal materials with high in-plane birefringence in our current research. We have carefully read this report and added this literature as ref [10] in the revised manuscript.

[Comment 3] *Is there a reason to cut off the data presented between 2.5 - 4 micron in the FTIR measurements?*

[Reply]

Thanks for pointing out this issue. Actually, we have the data between 2.5-4 micron in the FTIR measurements, as displayed in Figure 1c and 1e of the main text.

Figure 1c The FTIR experimental (unfilled symbols) and calculated (red line) reflectance spectra of a 5.58- μm -thick Ta_2NiS_5 sample on CaF_2 substrate. **e** Extracted complex refractive-index values of Ta_2NiS_5 for the a -, b -, and c -axis in the MIR range.

Perhaps, the reviewer may want to know why we cut off the data between 0.8-2.5 micron. This is a consequence of different measurement equipment. The optical responses in the visible and MIR region are measured by spectroscopic ellipsometry and FTIR, respectively. The measurement wavelength range is currently limited by the capability of our equipment.

[Actions]

On page 7 of the revised main text, we added the description “**The measurement wavelength range is currently limited by the capability of our equipment.**” in the figure caption of Figure 1.

[Comment 4] *As the authors have data on the dichroism (Δk), it may be worth including that in the manuscript and briefly discussing it.*

[Reply]

Thanks for the constructive suggestion. We have followed the reviewer’s suggestion and discussed the in-plane and out-of-plane dichroism of the Ta_2NiS_5 in the

revised manuscript.

In the visible range, the maximum in-plane $|\Delta k|$ is ~ 2.0 and the corresponding value for out-of-plane is ~ 2.7 . These values are much larger than other in-plane anisotropic 2D materials, such as GeSe (Δk of 0.90 at ~ 470 nm) [Yang, Y. et al. *In-Plane Optical Anisotropy of Low-Symmetry 2D GeSe*. Adv. Opt. Mater. 7, 1801311 (2019)], and ZrS₃ (Δk of 0.78 at ~ 500 nm) [Hou, S. et al. *Birefringence and Dichroism in Quasi-1D Transition Metal Trichalcogenides: Direct Experimental Investigation*. Small 17, 2100457 (2021)]. In the MIR range, the maximum in-plane and out-of-plane $|\Delta k|$ values are 0.357 and 0.48 at ~ 2.5 μm , respectively, which are among the highest reported dichroism in vdW crystals, to the best of our knowledge.

[Actions]

On page 9-10 of the revised SI, we added the **Supplementary Note 4: Giant dichroism of Ta₂NiS₅ nanosheet**.

We calculated the dichroism of Ta₂NiS₅, as shown in **Supplementary Figure 7**. In the visible range, the maximum in-plane $|\Delta k|$ is ~ 2.0 and the corresponding value for out-of-plane is ~ 2.7 . These values are much larger than other in-plane anisotropic 2D materials, such as GeSe (Δk of 0.90 at ~ 470 nm)¹⁶, and ZrS₃ (Δk of 0.78 at ~ 500 nm)¹⁷. In the MIR range, the maximum in-plane and out-of-plane $|\Delta k|$ values are 0.357 and 0.48 at ~ 2.5 μm , respectively, which are the highest reported dichroism among anisotropic crystals, to the best of our knowledge.

Supplementary Figure 7. The absolute dichroism ($|\Delta k|$) in **a** visible and **b** MIR spectral region.

On page 6 of the revised main text, we added the sentence “Meanwhile, the in-

plane dichroism of Ta_2NiS_5 also reaches a high value of ~ 2.0 in visible range, and 0.357 in the MIR range (Supplementary Note 4).”

[Comment 5] Did the authors perform DFT calculations to see if they can reproduce the experimental values in the theoretical models? Although polarizability of the species is a good argument, it is unclear whether that is a sufficiently rigorous argument.

[Reply]

We thank the reviewer for this constructive comment. To further verify the anisotropic optical properties of Ta_2NiS_5 , we have followed the reviewer’s suggestion and performed DFT calculations. We find that the DFT results can well match the experimental optical properties (see below Figure R6), further validating the main findings of the current manuscript.

Figure R6. Experimental and calculated optical properties of the Ta_2NiS_5 . a, b Experimental complex dielectric function along different crystal axis in the visible wavelength. c, d Experimental refractive and extinction in the visible and MIR wavelength along different crystal axis. e, f DFT calculated complex dielectric function along different crystal axis in the visible wavelength. g, h DFT calculated refractive and extinction in the visible and MIR wavelength along different crystal axis.

[Actions]

On page 6 of the revised main text, we added the sentence “To further verify the anisotropic optical properties of Ta₂NiS₅, we have also performed density functional theory (DFT) calculations to reproduce the experimental values (**Supplementary Note 5**). The DFT results can well match the experimental optical properties, further validating high optical anisotropy of Ta₂NiS₅.”

On page 15 of the revised main text, we added details about the DFT calculations in Methods section.

Computational methods. All calculations are performed by using the Vienna *ab initio* simulation package (VASP) with the projector-augmented wave (PAW) method⁶¹. The Perdew–Burke–Ernzerhof (PBE) exchange–correlation functional and van der Waals corrections of DFT-D3 method are used for the geometry optimization and electronic properties calculations of Ta₂NiS₅^{62,63}. GGA+U methods are applied to describe the strong correlation effects in the localized *d* states with 5 eV for Ta atoms and 6eV for Ni atoms, respectively⁶⁴. The plane-wave cutoff energy is set as 500 eV and the separation of k-sampling for the first Brillouin zone is set as 0.025 Å⁻¹. The convergence of force and total energy is set as 0.01 eV Å⁻¹ and 10⁻⁶ eV, respectively.

On page 10-11 of the revised SI, we added **Supplementary Note 5**: Theoretical calculation of the optical properties of Ta₂NiS₅.

We performed density functional theory (DFT) calculations to further study the anisotropic optical properties of Ta₂NiS₅. As shown in **Supplementary Figure 8**, the calculated optical properties including dielectric function, refractive and extinction match well with our experimental results. The shift of peaks relative to the experimental results is likely due to the inadequacy of Perdew-Burke-Ernzerhof (PBE) functional in describing the band gap and optical properties of semiconductors.

Supplementary Figure 8. Experimental and calculated optical properties of the Ta_2NiS_5 .

a, b Experimental complex dielectric function along different crystal axis in the visible wavelength. **c, d** Experimental refractive and extinction in the visible and MIR wavelength along different crystal axis. **e, f** DFT calculated complex dielectric function along different crystal axis in the visible wavelength. **g, h** DFT calculated refractive and extinction in the visible and MIR wavelength along different crystal axis.

[Comment 6] *What is the origin of the uptick in the k for a - and c - axes > 10 micron? Is it free carriers or phonons? It will be useful to know if there is any reported information on their electrical characteristics. Perhaps it can also help understand the SNOM results better.*

[Reply]

Thanks for raising this point. The uptick in the k for a - and c -axes in the wavelength longer than 10 microns is possibly induced by the free carrier absorption rather than phonon. We would like to elaborate on this phenomenon as below.

The absorption peaks of optical phonons are usually much sharper than the electronic absorption bands. This is not like the situation in our observation of the very broadband absorption in the 10-16 microns. Besides, the reported infrared spectra of Ta_2NiS_5 shows a phonon energy range below 50 meV (i.e., > 24.8 micron), [Larkin, T. I. et al. *Infrared phonon spectra of quasi-one-dimensional Ta_2NiSe_5 and Ta_2NiS_5* . Phys. Rev. B 98,

125113 (2018)], which is much longer than our observed range. The free carrier is another possible origin account for this absorption. In our research, a Drude-Lorentz model was applied to describe the optical response in the MIR region. The Drude part give the contribution of the free carrier to the absorption. In our Drude-Lorentz model, the fitted plasma frequencies from the Drude part are 1155.75 and 1545.28 cm^{-1} for *a*- and *c*-axis, respectively, which are close to the uptick of the MIR region. This gives us a clue to venture a guess that the absorption originates from free carriers.

However, we must emphasize that more rigorous analyses and experiments, possibly of low temperature or electrical characteristics, are needed to unambiguously uncover this problem, which is not the main focus of our current work.

[Actions]

On page 10 of the revised SI, we added a paragraph in **Supplementary Note 4: Giant dichroism of Ta₂NiS₅ nanosheet.**

It is notable an uptick in the *k* (**Figure 1e** in main text) for *a*- and *c*-axes in the wavelength longer than 10 microns occurs, which is possibly induced by the free carrier absorption rather than phonon. Since the absorption peaks of optical phonons are usually much sharper than the electronic absorption bands, this is not like the situation in our observation of the very broadband absorption in the 10-16 μm . Besides, the reported infrared spectra of Ta₂NiS₅ shows a phonon energy range below 50 meV (i.e., > 24.8 micron)¹⁸, which is much longer than our observed range. The free carrier is another possible origin account for this absorption. In our research, Drude-Lorentz model was applied to describe the optical response in the mid-infrared region, and the fitted plasma frequencies from the Drude part are 1155.75 and 1545.28 cm^{-1} for *a*- and *c*-axis, respectively, which are close to the uptick of the MIR region. This gives us a clue to venture a guess that the absorption originates from free carriers.

REVIEWERS' COMMENTS

Reviewer #1 (Remarks to the Author):

I thank the authors for their revisions according to my previous comments. The revised manuscript can be considered for publication in Nature Communications.

Reviewer #2 (Remarks to the Author):

The authors have addressed many of my concerns but I want to mention a follow up concern here.

I think the argument that layered materials are thin and hence, the light can travel through thin layers easily is not sufficient for birefringent materials. Birefringence is most meaningful if the light can travel in both the directions with different speed. The thickness will only help ameliorate the loss issue in one direction. I highly recommend that the authors remove the reference of light transmitting through thin layers as a way to circumvent losses in birefringent materials, but rather acknowledge the limitations it may pose on device design. This is precisely why I asked for the absorption edges along all the three directions. Even if this reduces the impact of the reported work, I think the current work is of interest to the community, as achieving large birefringence is a challenging goal. It will be disingenuous and misleading for the community to propagate this notion that the few layer thickness will overcome the loss issues for birefringent materials.

I recommend publication of the manuscript but this issue needs to be addressed in the manuscript.

Manuscript ID: NCOMMS-23-27157A

Title: Visible to Mid-infrared Giant In-Plane Optical Anisotropy in Ternary Van der Waals Crystals

Point-by-point responses to Reviewers' Comments

We sincerely thank the reviewers for the insightful comments again concerning our manuscript. These comments are very valuable for further improving the quality of this manuscript. The changes to the manuscript are indicated in **RED** in the marked version of the revised manuscript, and the point-by-point responses are listed below. We hope that the quality of our manuscript has met the publication standard after this revision.

Reviewer #2 (Remarks to the Author):

[Comment 1]

The authors have addressed many of my concerns but I want to mention a follow up concern here.

I think the argument that layered materials are thin and hence, the light can travel through thin layers easily is not sufficient for birefringent materials. Birefringence is most meaningful if the light can travel in both the directions with different speed. The thickness will only help ameliorate the loss issue in one direction. I highly recommend that the authors remove the reference of light transmitting through thin layers as a way to circumvent losses in birefringent materials, but rather acknowledge the limitations it may pose on device design. This is precisely why I asked for the absorption edges along all the three directions. Even if this reduces the impact of the reported work, I think the current work is of interest to the community, as achieving large birefringence is a challenging goal. It will be disingenuous and misleading for the community to propogate this notion that the few layer thickness will overcome the loss issues for birefringent materials.

I recommend publication of the manuscript but this issue needs to be addressed in the manuscript.

[Reply]

Thank the reviewer for the constructive suggestion. We deleted the sentence “Given the very thin nature of vdW materials, light within their absorption range can still transmit through these materials after a short transmission distance, thereby enabling their potential application of birefringent properties.” and added the sentence “However, one should also be aware of the non-negligible absorption induced losses in the visible spectrum may impose possible limitations on future device design.” in the revised main text.

[Actions]

On page 6 of the revised main text, we added the sentence “**However, one should also be aware of the non-negligible absorption induced losses in the visible spectrum may impose possible limitations on future device design.**”